# Nationwide screening for Fabry disease in unselected stroke patients

Aleš Tomek[1☯]*, Reková Petra[2☯], Jaroslava Paulasová Schwabová[1], Anna Olšerová[1], Miroslav Škorňa[3], Miroslava Nevšímalová[4], Libor Šimůnek[5], Roman Herzig[5], Štěpánka Fafejtová[6], Petr Mikulenka[7], Alena Táboříková[8], Jiří Neumann[8], Richard Brzezny[9], Helena Sobolová[10], Jan Bartoník[11], Daniel Václavík[12], Marta Vachová[13], Karel Bechyně[14], Hana Havlíková[15], Tomáš Prax[16], Daniel Šaňák[17], Irena Černíková[18], Iva Ondečková[19], Petr Procházka[20], Jan Rajner[21], Miroslav Škoda[22], Jan Novák[23], Ondřej Škoda[24], Michal Bar[25], Robert Mikulík[26], Gabriela Dostálová[27], Aleš Linhart[27], on behalf of the National Stroke Research Network, part of Czech Clinical Research Infrastructure Network (CZECRIN) and Czech Neurological Society, Cerebrovascular Section[¶]

1 Second Faculty of Medicine, Department of Neurology, Charles University and University Hospital Motol, Prague, Czech Republic, 2 First Faculty of Medicine, Department of Neurology and Center of Clinical Neuroscience, Charles University and General University Hospital, Prague, Czech Republic, 3 Department of Neurology, University Hospital Brno and Faculty of Medicine, Masaryk University, Brno, Czech Republic, 4 Department of Neurology, Hospital České Budějovice, České Budějovice, Czech Republic, 5 Faculty of Medicine in Hradec Králové and University Hospital Hradec Králové, Department of Neurology, Hradec Králové, Czech Republic, 6 Department of Neurology, Regional Hospital Karlovy Vary, Karlovy Vary, Czech Republic, 7 3rd Medical Faculty, Department of Neurology Neurology Dpt., Charles University and University Hospital Kralovské Vinohrady, Prague, Czech Republic, 8 Department of Neurology Neurology Dpt., Krajská zdravotní, a.s.—Hospital Chomutov, Chomutov, Czech Republic, 9 Department of Neurology Neurology Dpt., Regional Hospital Kladno, Kladno, Czech Republic, 10 Department of Neurology Neurology Dpt., Hospital Třinec, Třinec, Czech Republic, 11 Department of Neurology Neurology Dpt., Regional Hospital of Tomáš Baťa, Zlín, Czech Republic, 12 Department of Neurology, AGEL Research and Training Institute, Ostrava Vítkovice Hospital, Ostrava, Czech Republic, 13 Department of Neurology, Krajská zdravotní, a.s.—Hospital Teplice, Teplice, Czech Republic, 14 Department of Neurology, Hospital Písek, Písek, Czech Republic, 15 Department of Neurology, Regional Hospital Liberec, Liberec, Czech Republic, 16 Department of Neurology, Regional Hospital Pardubice, Pardubice, Czech Republic, 17 Department of Neurology, Palacký University Medical School and Hospital, Olomouc, Czech Republic, 18 Department of Neurology, Regional Hospital Kolín, Kolín, Czech Republic, 19 Department of Neurology, Krajská zdravotní, a.s.—Hospital Děčín, Děčín, Czech Republic, 20 Department of Neurology, Regional Hospital Uherské Hradiště, Uherské Hradiště, Czech Republic, 21 Department of Neurology, Municipal Hospital Ostrava, Ostrava, Czech Republic, 22 Department of Neurology, Regional Hospital Náchod, Náchod, Czech Republic, 23 Department of Neurology, Regional Hospital Česká Lípa, Česká Lípa, Czech Republic, 24 Department of Neurology, Hospital Jihlava, Jihlava, Czech Republic, 25 D epartment of Neurology, University Hospital Ostrava and Faculty of Medicine, Ostrava University, Ostrava, Czech Republic, 26 International Clinical Research Center and Department of Neurology, St. Anne's University Hospital and Medical Faculty of Masaryk University, Brno, Czech Republic, 27 First Faculty of Medicine, 2nd Department of Medicine–Department of Cardiovascular Medicine, Charles University and General University Hospital, Prague, Czech Republic

☯ These authors contributed equally to this work.
¶ Membership of the National Stroke Research Network is provided in the Acknowledgments.
* ales.tomek@gmail.com

## Abstract

### Background and aims

Fabry disease (FD) is a rare X-linked lysosomal storage disorder caused by disease-associated variants in the *alpha-galactosidase A* gene (*GLA*). FD is a known cause of stroke in



**Data Availability Statement:** All relevant data are available within the paper.

**Funding:** This received support from Takeda (formerly Shire), which awarded a grant to the

Czech Medical Society of J.E. Purkyně with primary investigator AT. The specific roles of these authors are articulated in the 'author contributions' section. The funders of grant had no role in study design, data collection and analysis, decision to publish, or preparation of the manuscript. No additional external funding was received for this study.

**Competing interests:** The authors have read the journal's policy and have the following competing interests: Takeda (formerly Shire) awarded a grant to the Czech Medical Society of J.E. Purkyně with primary investigator AT. AT, JPS, PR, GD, AL have received speaker's fees from Takeda. This does not alter our adherence to PLOS ONE policies on sharing data and materials. There are no patents, products in development or marketed products associated with this research to declare.

younger patients. There are limited data on prevalence of FD and stroke risk in unselected stroke patients.

## Methods

A prospective nationwide study including 35 (78%) of all 45 stroke centers and all consecutive stroke patients admitted during three months. Clinical data were collected in the RES-Q database. FD was diagnosed using dried blood spots in a stepwise manner: in males—enzymatic activity, globotriaosylsphingosine (lyso-Gb3) quantification, if positive followed by *GLA* gene sequencing; and in females *GLA* sequencing followed by lyso-Gb3.

## Results

986 consecutive patients (54% men, mean age 70 years) were included. Observed stroke type was ischemic 79%, transient ischemic attack (TIA) 14%, intracerebral hemorrhage (ICH) 7%, subarachnoid hemorrhage 1% and cerebral venous thrombosis 0.1%. Two (0.2%, 95% CI 0.02–0.7) patients had a pathogenic variant associated with the classical FD phenotype (c.1235_1236delCT and p.G325S). Another fourteen (1.4%, 95% CI 0.08–2.4) patients had a variant of *GLA* gene considered benign (9 with p.D313Y, one p.A143T, one p.R118C, one p.V199A, one p.R30K and one p.R38G). The index stroke in two carriers of disease-associated variant was ischemic lacunar. In 14 carriers of *GLA* gene variants 11 strokes were ischemic, two TIA, and one ICH. Patients with positive as compared to negative *GLA* gene screening were younger (mean 60±SD, min, max, vs 70±SD, min, max, P = 0.02), otherwise there were no differences in other baseline variables.

## Conclusions

The prevalence of FD in unselected adult patients with acute stroke is 0.2%. Both patients who had a pathogenic *GLA* gene variant were younger than 50 years. Our results support FD screening in patients that had a stroke event before 50 years of age.

## Introduction

Fabry disease (FD) is a progressive, X-linked inherited disorder of glycosphingolipid metabolism due to deficient or absent α-galactosidase-A (α-Gal-A) activity caused by over 1000 known disease-associated variants in the *GLA* gene [1]. The α-Gal-A deficiency results in the accumulation of globotriaosylceramide (Gb3) and other glycosphingolipids in various cell types, including capillary endothelial cells, renal, cardiac, and nerve cells [2].

FD is a disease with a broad spectrum of heterogeneously progressive clinical phenotypes due to the different residual levels of α-Gal-A activity. The most severe part of the spectrum is the classical phenotype in hemizygous males with onset in childhood or adolescence and symptoms including acroparesthesias, angiokeratomas, hypohidrosis, gastrointestinal symptoms, corneal dystrophy (cornea verticillata), cardiac, renal and cerebrovascular manifestations. On the other end of the spectrum are milder late-onset phenotypes observed in males with some degree of residual α-Gal-A activity and most symptomatic heterozygous females, with dominance of one organ system impairment, e.g., renal, cardiac, or nervous system including cerebrovascular disease [2].

The most devastating neurologic consequence of FD is stroke, which occurs at an increased prevalence and a younger age in the FD patients compared with the general population [3]. The prevalence of stroke in FD patients observed in the Fabry Outcome Survey was 11.1% in males and 15.7% in females [4]. Interestingly 50% of males and 38.3% of females experienced their first stroke before being diagnosed with Fabry disease [5]. The most frequent type of stroke in FD patients is ischemic (86.8%), but 16.9% of males and 6.9% of females had hemorrhagic strokes [5]. The small-vessel disease is the most commonly seen etiological subtype of ischemic stroke in young patients with FD, manifesting as either lacunar stroke or as asymptomatic cerebral white matter hyperintensity on magnetic resonance imaging. Nearly half of FD patients had small vessel disease on magnetic resonance imaging in one study [6]. The second most common ischemic stroke subtype in FD patients is cardioembolic stroke due to the higher incidence of left ventricular hypertrophy with diastolic function impairment, atrial fibrillation, prothrombotic state, and hypertension secondary to chronic renal failure [2,7,8]. Notably, late-onset patients are often misdiagnosed and unrecognized throughout life due to the atypical adult-onset and frequently silent family history. Therefore, many FD screening studies in the last twenty years concentrated on high-risk groups of patients–cardiac patients with cardiomyopathy, renal impairment patients on hemodialysis, and mostly younger cryptogenic ischemic stroke patients [9]. However, the data on the prevalence of *GLA* variants in stroke patients are sometimes conflicting since the pathogenicity of some of them had not been definitely established or are currently reclassified as benign [9,10]. A recent metanalysis of screening studies in stroke patients (5978 patients in total) found that 0.13% males and 0.14% females had a pathogenic *GLA* variant, and 0.54% males, and 0.96% females had a likely benign *GLA* variant [9]. Furthermore, the majority of screening studies have been done in younger patients with cryptogenic ischemic stroke and might not represent the true burden of FD pathology in unselected cerebrovascular disease populations.

## Aim and rationale

Considering the limited information available on the prevalence of FD in cerebrovascular disease outside the most studied group of young patients with cryptogenic ischemic stroke, we conducted the present study to investigate the prevalence of FD in unselected stroke patients irrespective of the stroke subtype, etiology, sex or age.

## Methods

### Study design and patient selection

We performed a prospective nationwide multicenter study, including consecutive stroke patients admitted during respective predefined three months in stroke centers in the Czech Republic. All consenting patients presenting with an acute cerebrovascular disease admitted during March 2018, October 2018, and March 2019 were included in the study irrespective of the exact stroke subtype. We included patients presenting during the study duration with a transient ischemic attack, ischemic stroke, intracerebral hemorrhage, subarachnoid hemorrhage, and cerebral venous thrombosis, regardless of stroke etiology or patient age. There were no other exclusion or inclusion criteria. All patients gave written informed consent with the study, including the genetic analyses. The study has a substudy with wider inclusion criteria that is still recruiting patients with cryptogenic ischemic stroke irrespective of age and date of the index stroke. The results of the of the substudy are not reported in this article.

## Data collection

Clinical data were collected in the Registry of Stroke Care Quality (RES-Q), an internet-based registry database used for continuous monitoring of the quality of stroke care by all certified stroke centers in the Czech Republic and the countries participating in European Stroke Organisation Enhancing and Accelerating Stroke Treatment (ESO—EAST) program [11]. Although RES-Q is primarily designed for monitoring the quality and especially the logistics of acute stroke care, we added a specific database entry on the inclusion in the study during the study periods. We collected demographic characteristics (age, gender), type of cerebrovascular event, stroke risk factors (hypertension, atrial fibrillation, carotid stenosis, previous stroke), initial stroke severity (NIHSS stroke scale), and clinical outcome at discharge (mRS scale). The full list of clinical parameters included in the RES-Q database was published in detail previously [11]. There was no predefined set of required clinical investigations. All patients were examined according to their treating physician adhering to the national standard of care [12].

## Diagnosis of Fabry disease

FD was diagnosed using dried blood spots in a stepwise manner combining genetic and enzyme testing. After obtaining the informed consent, peripheral blood was drawn and transferred to commercially available filtration paper (CentoCard, CentoGene AG). Samples were allowed to dry at room temperature, stored at a plastic sleeve in room temperature for no more than one week until sent for analysis. All samples of patients were analyzed in the designated study laboratory CentoGene AG (Rostock, Germany). The algorithm for FD screening was dependent on gender. In males, the enzymatic activity of alpha-galactosidase-A (α-Gal-A) and the concentration of the biomarker globotriaosylsphingosine (lyso-Gb3) were determined using fluorimetry and liquid chromatography-mass spectrometry, respectively [13]. If the activity of α-Gal-A was decreased < 15.3 μmol/L/h and/or the concentration of lysoGb3 was increased >1.8 ng/ml, then the *GLA* gene was sequenced. The *GLA* gene was analyzed by an amplicon-based next-generation sequencing approach. The amplicons covered the entire coding region and the highly conserved exon-intron splice junctions (used reference sequence of the *GLA* gene was NM_000169.2). The libraries were sequenced on an Illumina platform (MiSeq). An in-house bioinformatics pipeline including read alignment to GRCh37/hg19 genome assembly, variant calling and annotation is used. We have a minimum coverage of >20x for every amplicon. Missing regions or regions of poor quality are completed with classical Sanger sequencing to achieve 100% coverage. Given the expected high rate of false-negative results of enzymatic assays in females, *GLA* gene sequencing was the first method used. If a variant genotype was found, then the measurement of lyso-Gb3 was done. The result of FD screening was sent to the referring physician. Patients with positive screening results were referred from a stroke center to specialized Fabry Disease center in General Faculty Hospital, Prague, for further clinical and laboratory investigations and management, including family screening.

## Statistical analysis

Demographic and clinical data were summarized with descriptive statistics. Demographic parameters in patients with and without FD diagnosis were compared using Mann–Whitney U test for continuous variables and χ2 test or Fisher's exact test for non-continuous variables and p value was corrected for multiple comparisons. For comparison of genotype frequencies, we used Fisher's exact included in EPITAB function of STATA. P values <0.05 were considered statistically significant. Statistical analysis was performed using IBM SPSS Statistics 25 (IBM, USA) and STATA IC 16.1 (StataCorp, USA).

### Funding and study organization

The protocol was designed by the Executive Committee of the Cerebrovascular Section of the Czech Neurological Society and approved by the Executive Committee of the Czech Neurological Society. The Motol University Hospital ethics board approved the study protocol; subsequently, the protocol was approved by institutional boards in all participating sites. The Executive Committee of the Cerebrovascular section served as the Steering committee of the study. The required laboratory tests for FD were financed through a grant from Takeda (formerly Shire) to the Czech Neurological Society. The data collection and all other patient examinations excluding the specific FD tests were not reimbursed specifically outside the general health insurance. The sponsor of the study had no role in study design, data collection, analysis, or interpretation of the results.

## Results

We enrolled 986 consecutive acute cerebrovascular disease patients in 35 stroke centers. The detailed demographic and clinical parameters are given in **Table 1**. The mean age at stroke onset was 70.0 years (SD 12.8; range 24–97 years). Ischemic stroke was diagnosed in 79.6%, transient ischemic attack in 13.4%, intracerebral hemorrhage in 6.4%, subarachnoid hemorrhage in 0.5%, and cerebral venous thrombosis in 0.1% of our cohort. Sixteen (1.6%) patients

**Table 1. Demographic and clinical parameters of screened acute cerebrovascular disease patients.**

| | Entire cohort | Positive screening result—disease-associated variant /Variant in *GLA* gene present | Negative screening result—disease-associated variant /variant in *GLA* gene not present | P-value between groups[*] |
|---|---|---|---|---|
| Subjects, No. (%) | 986 | 16 (1.6) | 970 (98.4) | |
| Men, No. (%) | 536 (54.4) | 6 (37.5) | 530 (54.6) | 0.172 |
| Age, mean (SD), range, years | 70.0 (12.8), 24–97 | 60.1 (16.8), 34–83 | 70.2 (12.7), 24–97 | **0.021** |
| Ischemic stroke, No. (%) | 782 (79.3) | 13 (81.3) | 769 (79.3) | 0.845 |
| Transient ischemic attack, No. (%) | 134 (13.6) | 2 (12.5) | 132 (13.6) | 0.897 |
| Intracerebral hemorrhage, No. (%) | 64 (6.5) | 1 (6.3) | 63 (6.5) | 0.968 |
| Subarachnoidal hemorrhage, No. (%) | 5 (0.5) | 0 | 5 (0.5) | 0.685 |
| Cerebral venous thrombosis, No. (%) | 1 (0.1) | 0 | 1 (0.1) | 0.856 |
| Atrial fibrillation, No. (%) | 211 (21.4) | 2 (12.5) | 209 (21.5) | 0.545 |
| Hypertension, No. (%) | 771 (78.2) | 10 (62.5) | 761 (78.5) | 0.132 |
| Smoking, No. (%) | 256 (26.0) | 5 (31.3) | 251 (25.9) | 0.634 |
| Symptomatic internal carotid artery stenosis > 50% | 122 (12.4) | 3 (18.8) | 119 (12.3) | 0.435 |
| Recurrent stroke, No. (%) | 199 (20.2) | 2 (12.5) | 197 (20.3) | 0.489 |
| Onset NIHSS, mean (SD), range, points | 4.2 (5.2), 0–32 | 5.6 (4.5), 2–15 | 4.2 (5.1), 0–32 | 0.057 |
| Duration of initial hospitalization, mean (SD), days | 8.3 (8.8) | 6.1 (3.1) | 8.3 (8.9) | 0.687 |
| Good clinical outcome (discharge mRS 0–2) | 644 (67.2) | 14 (87.5) | 630 (66.9) | 0.107 |

NIHSS–National Institute of Health Stroke Scale, mRS–modified Rankin Scale.

[*]) Using the $\chi^2$ test, Fisher's exact test or Mann-Whitney U test as appropriate.

**Table 2. Basic characteristics of patients with disease-associated variant or variant genotype in *GLA* gene.**

| Patient, age [years] | GLA gene– nucleotide change | α-Gal-A– amino acid change | Lyso-Gb3 [ng/mL] | α-Gal-A activity [μmol/L/h] | Onset NIHSS | Stroke risk factors | Stroke type, etiology, territory | Pretsroke mRS | Discharge mRS |
|---|---|---|---|---|---|---|---|---|---|
| **Fabry disease associated *GLA* variants** | | | | | | | | | |
| Male, 34 | c.973G>A | p.G325S | 19.8 | <2.8 | 2 | Smoking | Ischemic, SVD, VB | 0 | 1 |
| Female, 40 | c.1235_1236delCT | **x** | n.a. | 6.22 | 3 | Smoking | Ischemic, SVD, VB | 0 | 0 |
| ***GLA* variant genotype with unclear significance or neutral to FD** | | | | | | | | | |
| Female, 83 | c.89G>A | p.R30K | 1.0 | n.a. | 2 | Hypertension, dementia | Ischemic, SVD, VB | 1 | 1 |
| Female, 78 | c.112A>G | p.R38G | 1.0 | n.a. | 8 | Atrial fibrillation, hypertension, PVD, previous ischemic stroke | Ischemic, cardioembolic, MCA occlusion | 1 | 1 |
| Female, 70 | c.427G>A | p.A143T | 1.1 | n.a. | 4 | Hypercholesterolemia, hypertension, PVD | TIA, MCA | 0 | 0 |
| Female, 47 | c.352C>T | p.R118C | 0.8 | n.a. | 3 | Hypertension | TIA, MCA | 0 | 0 |
| Male, 78 | c.596>C | p.V199A | 1.3 | 9.4 | 12 | Atrial fibrillation, IHD, smoking, COPD | Ischemic, cardioembolic, BA occlusion | 0 | 1 |
| Male, 41 | c.937G>T | p.D313Y | 1.6 | 11.1 | 15 | Hypertension | Intracerebral hemorrhage in basal ganglia | 1 | 4 |
| Female, 72 | c.937G>T | p.D313Y | 0.9 | n.a. | 3 | Hypertension, IHD | Ischemic, LVD, ICA | 1 | 1 |
| Male, 54 | c.937G>T | p.D313Y | 1.0 | 12.1 | 5 | Smoking, hypercholesterolemia | Ischemic, LVD, ICA | 0 | 2 |
| Male, 64 | c.937G>T | p.D313Y | 1.1 | 13.4 | 6 | Hypercholesterolemia, hypertension, IHD | Ischemic, SVD, VB | 0 | 4 |
| Female, 44 | c.937G>T | p.D313Y | 1.0 | n.a. | 15 | None | Ischemic, cryptogenic embolic, MCA | 0 | 1 |
| Female, 71 | c.937G>T | p.D313Y | 1.2 | n.a. | 4 | Hypercholesterolemia, hypertension | Ischemic, SVD, MCA | 0 | 1 |
| Female, 44 | c.937G>T | p.D313Y | 1.1 | n.a. | 2 | Previous ischemic stroke and TIA | Ischemic, SVD, MCA | 0 | 1 |
| Female, 83 | c.937G>T | p.D313Y | 1.4 | n.a. | 2 | DM, hypercholesterolemia, hypertension, IHD, PVD, pulmonary embolism | Ischemic, cryptogenic embolic, MCA | 2 | 2 |
| Male, 59 | c.937G>T | p.D313Y | 1.1 | 14.8 | 3 | Hypertension, smoking | Ischemic, SVD, MCA | 0 | 0 |

α-Gal-A = Alpha-galactosidase-A; lyso-Gb3 = globotriaosylsphingosine; n.a. = not available (α-Gal-A activity was not assessed in screening of females);

PVD = peripheral vascular disease; IHD = ischemic heart disease; DM = Diabetes mellitus; COPD = Chronic obstructive pulmonary disease; mRS = modified Rankin Scale; TIA = Transient ischemic attack; SVD = small vessel disease; LVD = large vessel disease; VB = vertebra-basilar territory; MCA = middle cerebral artery; ICA = Internal carotid artery; BA = basilar artery.

in total– 13 with ischemic stroke, 2 with TIA, and one intracerebral hemorrhage—had a positive result of the screening. Compared to the patients with negative screening, they were younger (mean 60.0, vs. 70.1, P = 0.02, range 34–83 years). There was a trend for fewer males (37.5% vs. 54.6%) and a higher admission NIHSS (5.6 vs. 4.2). Otherwise, there were not any significant differences observed in collected parameters. The clinical details on positively screened patients are given in **Table 2**.

## Two (0.2%) patients had a variant associated with the classical FD phenotype

Both patients and their families were undiagnosed with FD before the index stroke.

The first was male aged 34 years with lacunar thalamic infarction who had a disease associated *GLA* variant c.973G>A (G325S), decreased activity of α-Gal-A (<2.8 μmol/L/h) and increased LysoGb3 (19.8 ng/mL). After subsequent detailed screening for FD symptoms, incipient renal impairment and ocular pathology (marked tortuosity of the retinal vessels) were discovered, and patient was initiated on enzyme replacement therapy.

The second patient with pathogenic variant (heterozygous deletion c.1235_1236delCT) and normal α-Gal-A activity was female, aged 40 years, presenting with lacunar mesencephalic infarction. Five years before the index stroke, corneal opacities were noted in an ophthalmology exam, but the FD diagnosis was not followed through. Apart from ocular involvement, mild proteinuria and stroke, no other pathology was observed in the detailed screening of FD symptoms. The patient was also initiated on enzyme replacement therapy.

## Fourteen (1.4%) patients had genetic variants in the *GLA* gene that are mostly considered to be benign (FD neutral), or their clinical significance is not clear

The most prevalent variant genotype in nine patients was c.937G>T (D313Y). One of the carriers of p.D313Y was male, aged 41 years, with typical intracerebral hemorrhage in basal ganglia, all other patients had an ischemic stroke, details are given in **Table 2**. We found five different non-pathogenic variants in our cohort, each in one respective patient–female, aged 70 years c.427G>A (A143T); female, aged 47 years c.352C>T (R118C); male, old 78 years c.596>C (V199A); female, aged 83 years c.89G>A (R30K); and female, aged 78 years c.112A>G (R38G). If we exclude two patients with clearly disease associated variants, then the mean age of 14 patients with variant genotype was 63.4 (SD 15.2, range 41–83), statistically not different from patients with negative screening (P = 0.11).

The frequency of disease-assciated variants and variant of unclear clinical significance in *GLA* gene in the study population stratified by sex and age is shown in detail in **Table 3.** Variant genotype was the most frequent in females aged under 50 years (9.1%). On the other hand, the frequency of benign variant *GLA* genotype in the study population over 50 years was 1.4% in females and 0.8% in males. Irrespective of sex, we have found that in patients under 50 years of age, 2.5% of patients had a known disease-associated variant and that 5.1% of patients had a benign *GLA* variant genotype. We did not find any pathogenic disease-associated variant of the *GLA* gene in patients above the age of 50 years. We compared frequencies of benign *GLA* variants observed in our stroke cohort with available population data. We did not see any significant differences compared to the published frequencies of benign *GLA* variants in the general population of European descent. Details are given in **Table 4**.

**Table 3. Frequency of variant *GLA* genotype in study population stratified by sex and age.**

|  | Entire cohort | Females | Females 0–49 years | Females 50 + years | Males | Males 0–49 years | Males 50 + years |
|---|---|---|---|---|---|---|---|
| Subjects, No. (% of the entire cohort) | 986 (100) | 450 (45.6) | 33 (3.3) | 417 (42.4) | 536 (54.4) | 46 (4.7) | 490 (49.6) |
| Fabry disease associated variant carriers | 2 (0.2) | 1 (0.2) | 1 (3.0) | 0 | 1 (0.2) | 1 (2.2) | 0 |
| Variant considered benign (FD neutral or with unclear significance) | 14 (1.4) | 9 (2.0) | 3 (9.1) | 6 (1.4) | 5 (0.9) | 1 (2.2) | 4 (0.8) |

Number of subjects, percentages in parentheses are given according to the columns, except for the first row.

**Table 4. *GLA* variants considered benign or with unclear clinical significance discovered in our study and their population frequency.**

| Variant | dbSNP identificator | Patients with variant in our screened cohort (n = 986), No. | Allelic frequency of variant in screened cohort | General population allelic frequency | Sample size for population frequency, No. | P-value between groups using Fisher's exact test for comparison of proportions |
|---|---|---|---|---|---|---|
| c.89G>A (p. R30K) | Not included in dbSNP | 1 | 0.001 | 0.0000839[#] | n.a. [#] | 0.085 |
| c.112A>G (p. R38G) | rs730880446 | 1 | 0.001 | 0.0001399[#] | n.a. [#][*] | 0.128 |
| c.352C>T (p. R118C) | rs148158093 | 1 | 0.001 | 0.00052[*] | 79354[*] | 0.406 |
| c.427G>A (p. A143T) | rs104894845 | 1 | 0.001 | 0.00101[*] | 81358[*] | 1.00 |
| c.596T>C (p. V199A) | rs781871113 | 1 | 0.001 | 0.0002[*] | 6062[*] | 0.186 |
| c.937G>T (p. D313Y) | rs28935490 | 9 | 0.0091 | 0.00554[*] | 81344[*] | 0.130 |

Frequency data according to [*]) ALFA: Allele Frequency Aggregator." National Center for Biotechnology Information, US National Library of Medicine, European data (Phan et al., 2020)

[#]) CentoLSD database (https://www.centogene.com/centolsd.html) as available.

dbSNP = The Single Nucleotide Polymorphism Database of Nucleotide Sequence Variation (https://www.ncbi.nlm.nih.gov/snp/).

## Discussion

Our prospective countrywide screening study in unselected stroke patients found that 0.2% of patients had a disease-associated *GLA* variant with the classical FD phenotype. These numbers are comparable to the prevalence of 0.13% in males and 0.14% in females reported by a recent meta-analysis [9]. Similarly, we did not observe a different prevalence of likely benign *GLA* variants–in our study—0.9% in males and 1.4% in females compared to the meta-analysis— 0.54% in males and 0.96% in females [9].

The strength of our study compared to most of the previous studies is the inclusion of all consecutive stroke patients, irrespective of stroke subtype, stroke etiology, sex, and especially age. Our study included 907 patients above the age of 50 years. Actually, 25% of our cohort was older than 79 years. To our knowledge, only three studies up to date included stroke patients irrespective of their age. Two of them were done in Japan–Nakamura et al screened only males with mean age 69.7±12.5 years, and Nagamatsu et al tested both genders with mean age 74.1±12.5 years [14,15]. Marquardt et al. did the only study in stroke patients older than 60 years of age in the population of European descent with the mean age being 73.2 years, 85% patients in the cohort were older than 60 years [16]. That is fully comparable to our cohort, where 801 (81.2%) patients were older than 60 years. Compared to Marquardt et al. [16] we also included 64 patients with spontaneous intracerebral hemorrhage with mean age 69.3±13.7 years.

Clinically important finding is that FD with a disease-associated *GLA* variation is probably very rare or nonexistent in elderly stroke patients. Both our patients with a disease-associated variant were younger than 50 years. In fact, in this most frequently screened age subgroup, 2.5% of our cohort had a disease-associated *GLA* gene variant compared to none in patients above the age of 50 years. Therefore, based on our results we would not recommend screening patients suffering from stroke above 50 years of age.

The important finding of our study is the relatively high observed number of carriers of *GLA* variant p.D313Y because such finding may suggest connection between such variant and risk of stroke. Although our observed frequencies did not differ significantly from the largest

available populational data, we have seen a trend for a higher occurrence of p.D313Y in our cohort. We found 9 (0.9%) patients compared to 0.5% reported amongst populations of European descent [17]. The mean age of these nine patients, four males, and five females, was 59.1 years. None of them had elevated Lyso-Gb3. In four male carriers, where α-Gal-A activity was measured, none of them showed enzyme activity <35%. All of the nine patients did not have any apparent clinical signs of other organ involvement besides stroke; for further details, see **Table 2**.

The data regarding the pathogenicity and clinical relevance of the p.D313Y variant are despite more than two decades of research still controversial. Currently is the variant p.D313Y most often referred to as benign (International Fabry Disease Genotype-Phenotype Database, dbFGP, www.dbfgp.org) or polymorphism with unknown significance (http://fabry-database. org/). The carriers of the p.D313Y variant do not manifest the classical early-onset FD phenotype. The recent meta-analysis by Effraimidis et al. collected all current data about p.D313Y and concluded that carriers described in literature had high residual enzyme activity, low frequency of clinical features specific for FD, non-elevated lyso-Gb3/Gb3 concentrations and lack of intracellular Gb3 accumulation in skin and kidney biopsies [18]. The prevalence of the variant in populations with cardiac (0.20%) and renal (0.42%) disease was comparable to the reported frequency in the general population. A possible higher frequency was only observed in neurologic disorders–in stroke patients, 0.59%, and in small fiber neuropathy patients 0.80% [18]. Based on the published data and our findings, we think the possible higher frequency of p.D313Y variant in cerebrovascular disease might represent a low-effect stroke risk factor. If it is contributing to the overall stroke risk in its carriers, or it is an accidental finding, remains to be fully explained by further large populational studies.

Our study observed three more *GLA* variants each in one patient (0.1% of cohort), that are considered likely benign or again with unknown significance and were also reported in the published stroke screening studies—specifically, p.R118C, p.A143T, and p.V199A, for details on patients, see **Table 2**.

We also observed two variants that are reported for the first time in a stroke cohort or even in FD screening in general - c.89G>A (p.R30K) and c.112A>G (p.R38G). Both patients were elderly females with normal values of Lyso-Gb3, many other stroke risk factors, and no other symptoms specific for FD phenotype; details are given in **Table 2**. Based on this fact, we considered both variants benign, but at this point, we were not able to test them further.

Ours is also a first study that documents the occurrence of FD in stroke patients in the Czech Republic. In fact, it is the first such study in a population of central or east European descent. However, there were already two national screening studies in the Czech Republic in other typical high-risk groups. The highest prevalence was in patients with unexplained left ventricular hypertrophy, where 4 (4%) out of 100 screened males were diagnosed with FD [19]. In the screening study, which included 3370 hemodialyzed patients, 5 (0.15%) patients were diagnosed with FD [20]. It seems that the frequency observed in stroke patients (0.2%) was the second highest of the three risk groups.

Some limitations of our study are worth mentioning. First, although we aimed to enroll all consecutive patients, we could enroll only patients who were able to consent with inclusion into the study, so a selection bias towards less severe stroke cases cannot be excluded. In terms of testing, we used the sequencing of the *GLA* gene as the most reliable method for screening in all included females. But in males, we at first measured α-Gal-A activity and the biomarker Lyso-Gb3, then we sequenced the patients with abnormal values. Although this approach was used by many previous studies [21–23], it could have consequently underestimated the overall frequency, especially of benign variants in males, as some carriers probably had normal enzyme activity. Future studies should use the *GLA* sequencing also in male patients. Another

limitation of our approach to genetic testing was omitting to test for copy number variants. Our other shortcoming is the level of data details collected for the patients with negative screening. We utilized a stroke care quality database RES-Q. The RES-Q was not designed to collect an abundant amount of data, but it made our study feasible for a larger amount of participating stroke centers. We tried to address this limitation with a detailed data collection in positive cases.

## Conclusions

In conclusion, the prevalence of FD is relatively high in unselected Czech adult patients with acute stroke. However, both patients who had a disease-associated *GLA* gene variant were younger than 50 years at the time of stroke. Our results do not support further FD screening of elderly stroke patients in routine clinical practice. Because Fabry disease is a treatable condition, and the diagnosis has implications for other family members, stroke neurologists should be therefore aware of FD as a cause of stroke in younger age groups.

## Acknowledgments

We would like to thank the participating stroke centers and to all our patients and their families. We would like to thank to the rmembers of the National Stroke Research Network led by Robert Mikulík [Corresponding email ID: robert.mikulik@fnusa.cz]. We namely thank to the network members: Lenka Sobotková (Department of Neurology Neurology Dpt., Regional Hospital Kladno, Kladno), Zdeněk Topinka (Department of Neurology, Regional Hospital Kolín, Kolín, Czech Republic), Petr Geier (Department of Neurology, Regional Hospital Pardubice, Pardubice, Czech Republic), Jan Dienelt (Department of Neurology, Regional Hospital Liberec, Liberec, Czech Republic), Petr Suchomel (Department of Neurology, Regional Hospital Liberec, Liberec, Czech Republic), Svatopluk Ostrý (Department of Neurology, Hospital České Budějovice, České Budějovice, Czech Republic), Tereza Loučná (Department of Neurology, Second Faculty of Medicine, Charles University and University Hospital Motol, Prague), Petr Janský (Department of Neurology, Second Faculty of Medicine, Charles University and University Hospital Motol, Prague), Lubomír Jurák (Department of Neurology, Regional Hospital Liberec, Liberec, Czech Republic) for their contribution to the study.

## Author Contributions

**Conceptualization:** Aleš Tomek, Miroslava Nevšímalová, Roman Herzig, Jiří Neumann, Daniel Václavík, Daniel Šaňák, Ondřej Škoda, Michal Bar, Robert Mikulík, Aleš Linhart.

**Data curation:** Aleš Tomek, Anna Olšerová, Miroslav Škorňa, Miroslava Nevšímalová, Libor Šimůnek, Roman Herzig, Richard Brzezny, Helena Sobolová, Marta Vachová, Karel Bechyně, Hana Havlíková, Tomáš Prax, Daniel Šaňák, Irena Černíková, Iva Ondečková, Petr Procházka, Jan Rajner, Ondřej Škoda.

**Formal analysis:** Aleš Tomek, Jaroslava Paulasová Schwabová, Miroslava Nevšímalová, Petr Mikulenka, Jan Bartoník, Robert Mikulík.

**Funding acquisition:** Aleš Tomek, Ondřej Škoda, Robert Mikulík.

**Investigation:** Aleš Tomek, Reková Petra, Jaroslava Paulasová Schwabová, Anna Olšerová, Miroslav Škorňa, Miroslava Nevšímalová, Libor Šimůnek, Roman Herzig, Štěpánka Fafejtová, Petr Mikulenka, Alena Táboříková, Jiří Neumann, Richard Brzezny, Helena Sobolová, Jan Bartoník, Daniel Václavík, Marta Vachová, Karel Bechyně, Hana Havlíková, Tomáš Prax, Daniel Šaňák, Irena Černíková, Iva Ondečková, Petr Procházka, Jan Rajner, Miroslav

Škoda, Jan Novák, Ondřej Škoda, Michal Bar, Robert Mikulík, Gabriela Dostálová, Aleš Linhart.

**Methodology:** Aleš Tomek, Reková Petra, Jaroslava Paulasová Schwabová, Roman Herzig, Ondřej Škoda, Michal Bar, Robert Mikulík, Aleš Linhart.

**Project administration:** Aleš Tomek, Reková Petra, Jaroslava Paulasová Schwabová, Anna Olšerová, Miroslav Škorňa, Miroslava Nevšímalová, Libor Šimůnek, Roman Herzig, Štěpánka Fafejtová, Petr Mikulenka, Alena Táboříková, Jiří Neumann, Richard Brzezny, Helena Sobolová, Jan Bartoník, Daniel Václavík, Marta Vachová, Karel Bechyně, Hana Havlíková, Tomáš Prax, Daniel Šaňák, Irena Černíková, Iva Ondečková, Petr Procházka, Jan Rajner, Miroslav Škoda, Jan Novák, Ondřej Škoda, Michal Bar, Robert Mikulík, Gabriela Dostálová, Aleš Linhart.

**Resources:** Aleš Tomek, Daniel Šaňák, Michal Bar, Robert Mikulík, Aleš Linhart.

**Supervision:** Aleš Tomek, Michal Bar, Robert Mikulík.

**Validation:** Aleš Tomek, Gabriela Dostálová, Aleš Linhart.

**Writing – original draft:** Aleš Tomek.

**Writing – review & editing:** Aleš Tomek, Reková Petra, Jaroslava Paulasová Schwabová, Anna Olšerová, Miroslav Škorňa, Miroslava Nevšímalová, Roman Herzig, Štěpánka Fafejtová, Petr Mikulenka, Alena Táboříková, Jiří Neumann, Richard Brzezny, Helena Sobolová, Jan Bartoník, Daniel Václavík, Marta Vachová, Karel Bechyně, Hana Havlíková, Tomáš Prax, Daniel Šaňák, Irena Černíková, Iva Ondečková, Petr Procházka, Jan Rajner, Miroslav Škoda, Jan Novák, Ondřej Škoda, Michal Bar, Robert Mikulík, Gabriela Dostálová, Aleš Linhart.

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
