## [Decision Letter · Decision Letter 0]

19 May 2021

PONE-D-21-10014

Nationwide screening for Fabry disease in unselected stroke patients

PLOS ONE

Dear Dr. Tomek,

Thank you for submitting your manuscript to PLOS ONE. After careful consideration, we feel that it has merit but does not fully meet PLOS ONE’s publication criteria as it currently stands. Therefore, we invite you to submit a revised version of the manuscript that addresses the points raised during the review process.

Your paper was evaluated by an expert in clinical human genetics and myself. The topic is interesting, and the study design and data support the conclusion. However, several points need to be clarified. Please read the comment carefully and address the issues accordingly. 

We look forward to receiving your revised manuscript.

Kind regards,

Tomohiko Ai, M.D., Ph.D.

Academic Editor

PLOS ONE

Journal Requirements:

[AT, JPS, PR, GD, AL have received speaker’s fees from Takeda. The required laboratory tests for FD were financed through a grant from Takeda (formerly Shire) to the Czech Neurological Society. The funders had no role in study design, data collection and analysis, decision to publish, or preparation of the manuscript. There are no other relevant conflicts to report. The authors have no financial relationships pertinent to this article to disclose.]. 

We note that you received funding from a commercial source: Takeda (formerly Shire)

4. One of the noted authors is a group or consortium [National Stroke Research Network, part of Czech Clinical Research Infrastructure Network (CZECRIN) and Czech Neurological Society, Cerebrovascular Section]. In addition to naming the author group, please list the individual authors and affiliations within this group in the acknowledgments section of your manuscript. Please also indicate clearly a lead author for this group along with a contact email address.

Reviewers' comments:

Reviewer's Responses to Questions

**Comments to the Author**

1. Is the manuscript technically sound, and do the data support the conclusions?

Reviewer #1: Yes

2. Has the statistical analysis been performed appropriately and rigorously? 

Reviewer #1: I Don't Know

3. Have the authors made all data underlying the findings in their manuscript fully available?

Reviewer #1: Yes

4. Is the manuscript presented in an intelligible fashion and written in standard English?

Reviewer #1: Yes

5. Review Comments to the Author

Reviewer #1: The manuscript entitled: “Nationwide screening for Fabry disease in unselected stroke patients” presents an interesting analysis about the prevalence of Fabry disease in an unselected population of 986 consecutive individuals affected by stroke in the Czech Republic. The main findings are that, although rare, molecular signature leading to the diagnosis of FD was achieved in 0.2%, in patients younger than 50 y/o and that the pseudodefiency allele c.937G>T (p.Asp313Tyr) (aka D313Y) could represent a risk allele for increased stroke susceptibility.

The manuscript is clear, the experiments were carried appropriately, and the limitations carefully delineated. As the authors stated, although theirs is not the first study addressing FD prevalence in stroke patients, it appears to be the first in the Czech Republic.

There are few minor points the authors should address

• The authors should specify that the discriminant is not the age per se, but the age of onset of the stroke event/s. So, it would read better if they would specify that FD screening is appropriate for all subjects that had a stroke event before 50 years of age

• Given the limitation of the algorithm employing enzymatic activity and globotriaosylsphingosine (lyso-Gb3) quantification before genetic analysis in males, should they consider suggesting doing genetic testing for GLA in all subjects that had a stroke event before 50 years of age irrespective of biochemical analysis?

• In the Methods section, the authors stated that they used: “The GLA gene was analyzed by PCR and sequencing (NGS-Illumina) of the entire coding region and the highly conserved exon-intron splice junctions”. Could they provide more details about the platform and technical specifics used (i.e.: MiSeq, HiSeq, depth of coverage, bioinformatic analysis, etc.) and if copy number variants (CNV) have been analyzed. CNV are rare but can be identified in up to 5% of cases (see: https://www.ncbi.nlm.nih.gov/books/NBK1292/#fabry.Summary). If CNV was not done, please add to the limitations

• While the GLA variant c.1235_1236del (p.Thr412Serfs*) is an established pathogenic change (https://www.ncbi.nlm.nih.gov/clinvar/variation/198402/), for the c.973G>A (p.Gly325Ser) (aka G325S) in ClinVar there is conflicting classification (https://www.ncbi.nlm.nih.gov/clinvar/variation/92573/), If considered Pathogenic or Likely Pathogenic, please provide your classification and rationale using the ACMG 2015 guidelines (PMID: 25741868) and any professional updated guidelines

• Although still erroneously use in official publication and mistakenly tolerated by scientific journals, the official term is variant, not mutation (see: PMID: 25741868 under “Terminology”)

• The authors should specify that the c.937G>T (p.Asp313Tyr) (aka D313Y) is a common pseduodeficiency allele that does not cause disease, but individuals with this variant can exhibit low alpha-galactosidase activity during enzyme analysis. As far as the other variants, the authors correctly identified them as potentially benign or variants of uncertain significance (VUS). In the case of the two novel GLA variants, p.Arg30Lys and p.Arg38Gly, although the age at testing (please provide the age at stroke event onset, if known) is favoring a more tolerated role, the authors should discuss the fact that, most females (not only with regard to those two variants, but in general), harboring a GLA variant, may or may not present the whole collection of symptoms, due to absolute gene dosage (heterozygous versus hemizygous in males) and the phenomenon of X-inactivation

• Was all the participant ancestry background uniform, or were there a significant ancestry background admixture? Some variants may be common in population isolates due to founder effect

• Please use the most current human genome variation nomenclature (https://varnomen.hgvs.org/)

6. PLOS authors have the option to publish the peer review history of their article (what does this mean?). If published, this will include your full peer review and any attached files.

Reviewer #1: **Yes: **Matteo Vatta

---

## [Author Response · Author response to Decision Letter 0]

23 Aug 2021

We have added all requested answers to the uploaded document "response to the reviewers", we have updated the cover letter as requested.

Thanks again for reviewing our submission!

---

## [Decision Letter · Decision Letter 1]

15 Nov 2021

Nationwide screening for Fabry disease in unselected stroke patients

PONE-D-21-10014R1

Dear Dr. Tomek,

We’re pleased to inform you that your manuscript has been judged scientifically suitable for publication and will be formally accepted for publication once it meets all outstanding technical requirements.

Kind regards,

Tomohiko Ai, M.D., Ph.D.

Academic Editor

PLOS ONE

Additional Editor Comments (optional):

Reviewers' comments:

Reviewer's Responses to Questions

**Comments to the Author**

1. If the authors have adequately addressed your comments raised in a previous round of review and you feel that this manuscript is now acceptable for publication, you may indicate that here to bypass the “Comments to the Author” section, enter your conflict of interest statement in the “Confidential to Editor” section, and submit your "Accept" recommendation.

Reviewer #1: All comments have been addressed

2. Is the manuscript technically sound, and do the data support the conclusions?

Reviewer #1: Yes

3. Has the statistical analysis been performed appropriately and rigorously? 

Reviewer #1: N/A

4. Have the authors made all data underlying the findings in their manuscript fully available?

Reviewer #1: Yes

5. Is the manuscript presented in an intelligible fashion and written in standard English?

Reviewer #1: Yes

6. Review Comments to the Author

Reviewer #1: The authors have adequately answered to all reviewer's comments and the manuscript's overall quality was improved.

7. PLOS authors have the option to publish the peer review history of their article (what does this mean?). If published, this will include your full peer review and any attached files.

Reviewer #1: No

---

## [Editor Report · Acceptance letter]

6 Dec 2021

PONE-D-21-10014R1 

Nationwide screening for Fabry disease in unselected stroke patients 

Dear Dr. Tomek:

I'm pleased to inform you that your manuscript has been deemed suitable for publication in PLOS ONE. Congratulations! Your manuscript is now with our production department. 

Kind regards, 

on behalf of

Dr. Tomohiko Ai 

Academic Editor

PLOS ONE